# MAGI1 Mediates eNOS Activation and NO Production in Endothelial Cells in Response to Fluid Shear Stress

**DOI:** 10.3390/cells8050388

**Published:** 2019-04-27

**Authors:** Kedar Ghimire, Jelena Zaric, Begoña Alday-Parejo, Jochen Seebach, Mélanie Bousquenaud, Jimmy Stalin, Grégory Bieler, Hans-Joachim Schnittler, Curzio Rüegg

**Affiliations:** 1Pathology, Department of Oncology, Microbiology and Immunology, Section of Medicine, Faculty of Science and Medicine, University of Fribourg, Chemin du Musée 18, CH-1700 Fribourg, Switzerland; kedar@kedarghimire.com (K.G.); jelena.zaric@epfl.ch (J.Z.); begona.aldayparejo@unifr.ch (B.A.-P.); melanie.bousquenaud@unifr.ch (M.B.); jimmy.stalin@unifr.ch (J.S.); gregory.bieler@unifr.ch (G.B.); 2Institute of Anatomy and Vascular Biology, Westfälische, Wilhelms-Universität Münster, Vesaliusweg 2-4, D-48149 Münster, Germany; seebach@uni-muenster.de (J.S.); hans.schnittler@uni-muenster.de (H.-J.S.); 3Cells-in-Motion Cluster of Excellence (EXC 1003—CiM), University of Muenster, D-48149 Muenster, Germany

**Keywords:** fluid shear stress, MAGI1, signal transduction, eNOS, endothelial cell, nitric oxide/NO

## Abstract

Fluid shear stress stimulates endothelial nitric oxide synthase (eNOS) activation and nitric oxide (NO) production through multiple kinases, including protein kinase A (PKA), AMP-activated protein kinase (AMPK), AKT and Ca^2+^/calmodulin-dependent protein kinase II (CaMKII). Membrane-associated guanylate kinase (MAGUK) with inverted domain structure-1 (MAGI1) is an adaptor protein that stabilizes epithelial and endothelial cell-cell contacts. The aim of this study was to assess the unknown role of endothelial cell MAGI1 in response to fluid shear stress. We show constitutive expression and co-localization of MAGI1 with vascular endothelial cadherin (VE-cadherin) in endothelial cells at cellular junctions under static and laminar flow conditions. Fluid shear stress increases MAGI1 expression. MAGI1 silencing perturbed flow-dependent responses, specifically, Krüppel-like factor 4 (KLF4) expression, endothelial cell alignment, eNOS phosphorylation and NO production. MAGI1 overexpression had opposite effects and induced phosphorylation of PKA, AMPK, and CAMKII. Pharmacological inhibition of PKA and AMPK prevented MAGI1-mediated eNOS phosphorylation. Consistently, MAGI1 silencing and PKA inhibition suppressed the flow-induced NO production. Endothelial cell-specific transgenic expression of MAGI1 induced PKA and eNOS phosphorylation in vivo and increased NO production ex vivo in isolated endothelial cells. In conclusion, we have identified endothelial cell MAGI1 as a previously unrecognized mediator of fluid shear stress-induced and PKA/AMPK dependent eNOS activation and NO production.

## 1. Introduction

Nitric oxide (NO) is a key regulator of homeostasis and adaptive responses of the vascular system [1]. NO is generated from L-arginine and molecular oxygen by a family of enzymes called nitric oxide synthases (NOS). There are three mammalian NOS isoforms: neuronal NOS (nNOS, NOS I), inducible NOS (iNOS, NOS II) and endothelial NOS (eNOS NOS III) [1]. They consist of an *N*-terminal oxygenase domain with binding sites for heme, l-arginine and tetrahydrobiopterin; a central calmodulin-binding region and a C-terminal reductase domain containing binding sites for nicotinamide adenine dinucleotide phosphate, flavin adenine dinucleotide and mononucleotide [2]. The main NOS isoform expressed in endothelial cells is eNOS, which acts as an important regulator of essential vascular functions including vascular tone, angiogenesis, atheroprotection, anti-inflammatory and anti-thrombotic activities [3,4]. A reduction of endothelial NO production is associated with endothelial dysfunctions as observed in hypertension, diabetes and atherosclerosis [5,6]. Regulation of eNOS is complex and occurs at transcriptional, post-transcriptional and post-translational levels [7,8]. Fluid shear stress and cyclic stretch are important inducers of eNOS expression and activity [9]. A main mechanism of regulation of eNOS activity is phosphorylation [4]. Several tyrosine (Tyr), serine (Ser) and threonine (Thr) residues are phosphorylated on eNOS. Ser^1177^ phosphorylation is critical for its activation and function [10] and can be mediated by AKT [11], AMP-activated protein kinase (AMPK) [12], Ca^2+^/calmodulin-dependent protein kinase II (CaMKII) [13] and protein kinase A (PKA) [14]. AKT and PKA were shown to be the main mediators of Ser^1177^ phosphorylation in response to shear stress. The roles of other phosphorylation sites, in particular Tyr^8^, Ser^114^, Ser^633^ and Ty^r657^, are still under investigation [1,4].

MAGI1 (MAGUK with inverted domain structure-1), together with MAGI2 and MAGI3, is a member of the MAGUK family of proteins. It consists of six PDZ (post-synaptic density protein 95 (PSD-95)-Drosophila disc large tumour suppressor (Dlg1)-zona occludens 1 (ZO-1)) domains, a single catalytically inactive guanylate kinase domain and two WW (tryptophan-tryptophan) domains [15]. MAGI1 exerts a scaffolding function supporting the assembly of multiprotein complexes at the inner surface of the plasma membrane, particularly at regions of cell-cell contacts thereby stabilizing cell-cell adhesions [16]. In epithelial cells, MAGI1 localizes at cellular junctions and recruits the tumour suppressor PTEN (phosphatase and tensin homolog) to stabilize adherens and tight junctions and to suppress cell motility, the latter, at least in part, by competing with TRIP6 (thyroid receptor-interacting protein 6) binding to PTEN [16,17].

Little is known about the role of MAGI1 in endothelial cell functions particularly its response to shear stress. MAGI1 was shown to mediate Rap1 activation and enhancement of VE-cadherin-dependent endothelial cell-cell adhesion [18]. Endothelial cell-selective adhesion molecule (ESAM) binds MAGI1 and recruits it to cell-cell contacts to strengthen adhesions [19]. Here, we report MAGI1 as a novel fluid shear stress-induced transcript/protein in endothelial cells that regulates eNOS phosphorylation and NO production through PKA/AMPK-dependent phosphorylation at Ser^1177^.

## 2. Materials and Methods

### 2.1. Reagents

Dulbecco PBS (DPBS), Ca^2+^-free PBS, medium-199 (M199), DMEM, penicillin/streptomycin, G418 and trypsin were purchased from Invitrogen Life Technologies (Basel, Switzerland). KAPA SYBR FAST Universal 2× quantitative PCR Master Mix kit was from Kapa Biosystems (Boston, MA, USA). Foetal bovine serum (FBS) was from Eurobio (Les Ulis, France). Luminata Western HRP substrate was purchased from Millipore (Boston, MA, USA). DAPI (4,6-diamidino-2-phenylindole), BSA, bovine gelatine, plasma fibronectin, collagen-I, PFA, crystal violet, SigmaFAST protease inhibitors, thrombin and puromycin were purchased from Sigma-Aldrich (Buchs, Switzerland). Human VEGF-A 165 was from R and D Systems (Minneapolis, MN, USA). Bradford and RC-DC assays for protein concentration measurement were from Bio-Rad (Richmond, CA, USA). PageRuler prestained and Spectra multicolour protein ladder were from Thermo Fisher Scientific (Reinach, Switzerland). Diaminofluorescein-2 Diacetate (DAF-2DA) was from Cell Technology Inc. (San Jose, CA, USA), Cell Meter™ Fluorimetric Intracellular Nitric Oxide (NO) Activity Assay Kit *Red Fluorescence Optimized for Flow Cytometry (cat. no. 16356) is from AAT Bioquest, Inc. (Sunnyvale, CA, USA). AKT inhibitor MK-2206, CAMKII inhibitor KN-93 phosphate and AMPK specific inhibitor dorsomorphin (compound C) were purchased from Selleck Chemicals (Huston, TX, USA). The PKA inhibitor, H-89 was purchased from Cell Signalling Technology (Danvers, MA, USA).

### 2.2. Antibodies

Antibodies to MAGI1 (rabbit polyclonal (cat. no. M5691) Prestige, cat. no. HPA031853, mouse monoclonal antibody, cat. no. WH0009223M3), actin (cat. no. A3853) and GAPDH (cat. no. G9545) were from Sigma-Aldrich (Buchs, Switzerland). Anti-CAMKII (phospho Thr286, cat. no. 3361 and total, cat. no. 3362), anti-PKA (phospho Thr197, cat. no. 4781 and total, cat. no. 4782), anti-AKT (phospho, Ser473, cat. no. 9271 and total cat. no. 9272), anti-KLF4 (cat. no. 4038) and anti-eNOS (phospho Ser1177 cat. no. 9572 and total, cat. no. 9571), anti-AMPKα (phospho Thr172, cat. no. 2535, total cat. no. 2532) antibodies were from Cell Signalling Technology (Danvers, MA, USA). Anti-VE-cadherin antibody (cat. no. 605252) was from BD Biosciences (Allschwill, Switzerland). Phalloidin–FITC (cat. no. P5282) was from Sigma-Aldrich (Buchs, Switzerland). Goat anti-mouse IgG–HRP (cat. no. P0447) and goat anti-rabbit IgG-HRP (cat. no. P0448) were from DAKO (Glostrup, Denmark). Fluorochrome-conjugated anti-mouse (cat. no. A10037) and anti-rabbit (cat. no. A21206) antibodies for immunofluorescence were purchased from Invitrogen, Thermo Fisher Scientific (Waltham, MA, USA). Florescent secondary antibodies for Western blotting were: goat anti-mouse IgG or anti-rabbit IgG Dylight 680 or 800 from Thermo Fisher Scientific (Waltham, MA, USA).

### 2.3. Cell Culture

HUVEC (human umbilical vein endothelial cells) were prepared from umbilical cords and cultured as previously described [20] and used between passage 3 and 7. HEK293T cells were grown in DMEM supplemented with 10% FBS and 1% penicillin/streptomycin. Use of HUVEC was approved by the both ethic committees of Canton Vaud, Switzerland and WWU–Muenster (2009-537-f-S). HUVEC stimulation with: VEGF: 100 ng/mL for 24 h; thrombin: 10 U/mL for 30 min.

### 2.4. Immunofluorescence Staining and Confocal Microscopy

HUVECs were seeded on gelatin-coated glass coverslips in a 12-well plate or IBIDI slides (ibidi, Martinsried, Germany) and cultured to confluency. Upon treatment, cells were washed with DPBS, fixed in PFA for 5 min at room temperature, blocked and permeabilized (0.5% BSA, 5% donkey serum, 0.1% TritonX-100 in DPBS) for 30 min. Cells were incubated with primary antibodies overnight at 4 °C, washed with DPBS 4–5 times and incubated with relevant fluorescent-conjugated secondary antibodies. DAPI (4,6-diamidino-2-phenylindole) was used to counterstain nuclei. The stained cells were mounted with ‘Prolong’ antifade mounting medium (Molecular Probes (Basel, Switzerland) and imaged using a Leica TCS SP5 DMI6000 confocal microscope (Leica Microsystems, Wetzlar, Germany).

### 2.5. Plasmids, Lentivirus Production and Transduction

For gene silencing experiments, the pCMVGIN- ZEO lentiviral shRNAmir expressing system was purchased from Open Biosystems (Huntsville, AL, USA). For the shRNAmir sequences directed against the human MAGI1, three different clones were used: V2LHS-36239 (SH1), V2LHS-36236 (SH2) and V3LHS_346390 (SH4). SH1 and SH2 shRNAmir sequences and a non-silencing control shRNAmir sequence that contains no homology to known mammalian genes, were subcloned into the pCMV-GIN-ZEO lentiviral system. SH4 was used in the original vector. Human MAGI1 cDNA was kindly provided by Dr Y. Hata (Dept. Medical Biochemistry, Graduate School of Medicine, Tokyo Medical and Dental University. Tokyo, Japan) and sub-cloned into pRRLSIN.cPPT.PGK/GFP.WPRE lentiviral vector under the control of the PGK promoter (= pSD44 plasmid). Cell cultures were transduced by over-night incubation at 37 °C in virus-containing media in the presence of 8 μg/mL polybrene (Sigma-Aldrich, Buchs, Switzerland). Selection was started on bulk cultures 48 h after transduction using 2 μg/mL Puromycin (Sigma-Aldrich) for pSD44-MAGI1-cDNA construct or 250 μg/mL G418 (Calibiochem, CA, USA) for pCMV-GIN-ZEO constructs.

### 2.6. Fluid Shear Stress Experiments

*Ibidi parallel plate flow system*- The protocol was based on the original method [21]. ‘Ibitreat’ slides were coated with 1% gelatin for 4 h at 37 °C. Two types of channel slides were used: μ-Slide I 0.4 luer and μ-Slide I 0.6 luer for flow applications and high channel (0.8 luer) for static conditions. For μ-Slide I 0.4 luer (100 μL channel volume); 1 × 10^6^ cells/mL were plated while 1.6 × 10^6^ cells/mL were plated in μ-Slide I 0.6 luer (150 μL channel volume) to obtain a 100% confluent cell layer after 24 h. Confluent slides were placed under flow using ibidi pump system (ibidi, Martinsried, Germany) at 10 dy/cm^2^ steady, unidirectional shear stress for 24 h.

*Orbital-shaker system*- The protocol modified form the original method [20]. 6-well plates were coated with 1% gelatin, 3 μg/mL fibronectin or 1 μg/mL collagen for 4 h. HUVEC were plated and cultured to confluence. Confluent plates were fixed on a horizontal orbital shaker (Vibromix 314 EVT orbital shaker, Tehtnica, Železniki, Slovenia) with an orbit of 38 mm generating a modelled oscillatory shear stress of 10–13 dyn/cm^2^ at the periphery of the plate as previously reported using the finite element method [20]. After 24 h, the cells at the central half of the diameter (exposed to a modelled shear stress < 5 dyn/cm^2^) were removed with cell scraper, leaving the periphery of the cultures intact. Cells were imaged using a Leica AF6000 microscope (Leica Microsystems, Wetzlar, Germany). For Western blotting, RIPA lysis buffer was added to the wells and processed as described [22].

*Cone-and-plate system*- Experiments were conducted using the fully automated BioTechFlow system (BTF), (MOS Technologies, Telgte, Germany) as described elsewhere [23,24]. The chamber plates were coated with 0.5% crosslinked gelatine as described elsewhere (25) before cells were seeded at 30,000 cells/cm^2^ in endothelial cell growth medium (Promocell, Heidelberg, Germany) under 5% CO_2_, 50 U/mL/50 μg/mL of Penicillin-Streptomycin, 3% Polyvinyl-pyrrolidone (MW: 360 kDa) was added to increase the dynamic viscosity. Diameter of the cone was 28 mm and the cone angle was 2.5°. Cells were exposed to 10 dyn/cm^2^ shear stress for 24 h. Phase-contrast microscopy is integrated to the system allowing real-time observation of endothelial cells.

### 2.7. Real-Time PCR

Total RNA was extraction, reverse transcription and real-time PCR were done as reported earlier [22]. Ct values of target genes normalized upon the Ct values of housekeeping genes (GAPDH for HUVEC and mouse 36B4 for mouse lung endothelial cells) were calculated between samples and the expression level was compared. All real time PCR reactions were done in triplicate, and the data from one of three independent experiments are shown. Primer sequences used were as follows: MAGI1 (NM_004742.2) forward: 5′-TTCAAGCCGTCAGACAAG-3′, MAGI1 reverse: 5′-ATGGCCGTAAAGGTTATCCC-3′; GAPDH forward: 5′-GGACCTGACCTGCCGTCTAG-3′; GAPDH reverse: 5′-CCACCACCCTGTTGCTGTAG-3′, Mouse 36B4: Forward: 5′-GTGTGTCTGCAGATCGGGTAC-3′. Reverse: 5′-CAGATGGATCAGCCAGGAAG-3′.

### 2.8. Western Blotting

SDS PAGE and Western blotting were performed following standard procedures as described [22]. Each sample (=lane) consists of the pool of three independent culture wells of a six-well plate. Detection was performed by chemo luminescence using Luminata Substrate or by Li-cor Odyssey CLx (Lincoln, NE, USA) using florescent secondary antibodies diluted in 1% BSA and blocking buffer for fluorescent antibody (Odyssey blocking buffer (TBS) from Li-cor). For quantification Western blot protein densitometry data were quantified with ImageJ and Alpha View software (Biotechne, San Jose, US) based on multiple experiments. 

### 2.9. NO Measurement

*The Cell Meter™ Fluorimetric Intracellular Nitric oxide (NO) Activity Assay Kit*- Trypsinised HUVECs were resuspended in medium at 5 × 10^5^ cells/mL and incubated with 2 µL of 500× Nitrixyte™ Red (Component A) for 30 min at 37 °C. After washing in PBS, the cells were monitored for the fluorescence intensity at the FL4 channel (Ex/Em = 630/660 nm) using a FACSCalibur flow cytometer (BD, Mountain View, CA). The negative controls were cells without the Nitrixyte™ Red probe added. The data obtained were analysed by FlowJo software (Ashland, OR, USA). 

*The DAF-2DA fluorescence labelling technique*- HUVEC cultures at confluence were washed with PBS and incubated with 5 µmol/L of DAF-2DA for 30 min. After incubation, cells were washed twice with PBS, and fixed with 2.5% PFA. Pictures were taken with an inverted fluorescent microscope (Leica DMI6000B, Leica Microsystems, Wetzlar, Germany) and intensity values were obtained with emission at 515 nm with a VICTOR X3 spectrophotometer from PerkinElmer Inc. (Waltham, MA, USA). 

### 2.10. Kinase Inhibition Assays

Confluent HUVECs were serum-starved for 2 h in M199 medium with 0.5% FCS except for CaMKII inhibition. Inhibitors were dissolved in DMSO at 10 to 50 mM and diluted in M199 medium immediately before use. Cells were then treated as following: MK-2206, 1 μM, for 1 h; H-89, 20 μM, 30 min; Dorsomorphin, 10 µM for 2 h no serum starvation; KN-93, 20 μM, 24 h, no serum starvation.

### 2.11. Animal Studies

The human MAGI1 cDNA (NM_004742.2, 3770bp) was inserted into the *pTetOS* responder construct to generate transgenic animals. Driver and responder transgenic animals were bred to generate bigenic mice. Offspring was genotyped to generate wild type, single (ST, tetOS:MAGI1 and VEC:tTA) and double transgenics (DT, VEC:tTA:: tetOS:MAGI1) mice. In the absence of doxycycline, mice constitutively overexpress transgenic MAGI1 in endothelial cells while in the presence of doxycycline transgenic expression of MAGI1 is silenced. Doxycycline treatment involved the addition of 100 μg/mL of doxycycline (cat. no. D9891, Sigma-Aldrich)]/5% sucrose in the drinking water and was changed at least twice per week. Animals were euthanized by CO_2_ inhalation followed by neck dislocation. Animal experiments were approved by the Cantonal Office in Fribourg (Ruegg_2014_26_FR) and performed according to Swiss regulations and to the guidelines from Directive 2010/63/EU of the European Parliament on the protection of animals used for scientific purposes. We used both male and female mice between 6 and 10 weeks of age. The following primers were used for genotyping the mice: VEC_forward: 5′GACGCCTTAGCCATTGAGAT 3′, VEC_reverse: 5′CAGTAG TAG GTGTTTCCCTTTCTT 3′, MAGI1_forward: 5′ TCATTCCTGGGCATGAGTCCT 3′, MAGI1_reverse: 5′GCCAGGGAAGGAAGGATTGT3′.

### 2.12. Isolation of Mouse Lung Endothelial Cells

Lungs from freshly sacrificed mice were cut and digested in 1% Collagenase and 2.5 µg/mL of DNase I, both from Sigma-Aldrich (Buchs, Switzerland) for 45 min at 37 °C. After passing through 70 µm filters, cells were washed once in PBS with 2 mM EDTA and twice in PBS only. Cells were resuspended in DMEM:F12 supplemented with 2% FBS, 1% penicillin/streptomycin, 20 ng/mL EGF and 10 µg/mL insulin and plated on Collagen I (10 µg/mL) coated plates.

### 2.13. Immunohistochemical Staining

Tissue sections were heated in Tris-EDTA buffer to retrieve antigen epitopes, blocked by 10% normal goat serum and Avidin/Biotin blocking reagent (Vector Laboratories, Burlingame, CA, USA) and stained with the following primary antibodies at 4 °C overnight: anti-MAGI1 (Sigma-Aldrich, Buchs, Switzerland, cat. no. HPA031853), P-eNOS (Ser1177, Cell Signaling, Danvers, MA, USA; cat. no. 9571S) and total eNOS (Cell Signaling, Danvers, MA, USA; cat. no. 9572S). Sections were incubated with biotinylated secondary antibodies followed by Vectastain ABC Kit (Vector Laboratories, Burlingame, CA, USA) with DAB peroxidase substrate (Sigma-Aldrich). Sections were counterstained with haematoxylin before mounting.

### 2.14. Statistical Analysis

Statistical analysis on data expressed as the mean ± SEM was performed on the basis of unpaired *t*-test or two-way ANOVA, followed by Sidak’s multiple comparison test, where necessary using GraphPad Prism software (San Diego, CA, USA). A value of *p* < 0.05 was considered to be significant. **p* < 0.05; ** *p* < 0.01; *** *p* < 0.001; **** *p* < 0.0001. N, repeated experiments; n, replicates per experiment. 

## 3. Results

### 3.1. MAGI1 Localizes at Endothelial Cell-Cell Contacts and its Expression is Induced by Fluid Shear Stress

The role of MAGI1 in vascular biology and its response to fluid shear stress are largely unknown. To address this question, we first monitored MAGI1 expression and localization in confluent human umbilical vein endothelial cells (HUVEC) by confocal immunofluorescence microscopy. Under static conditions (0.5 dyn/cm^2^), MAGI1 localized at cell-cell contacts as continuous staining in co-localization with VE-cadherin (Figure 1A), consistent with previous reports [18]. Upon 24 h exposure to fluid shear stress (10 dyn/cm^2^) in the cone-and-plate BioTechFlow-system (BTF) [25], we observed HUVEC alignment and a linear but more interdigitated VE-cadherin localization, as a sign of response to flow consistent with previous reports [25,26] and MAGI1 co-localized with VE-cadherin at cell-cell junctions (Figure 1A, Appendix A).

Next, we monitored the effect of shear stress (10 dyn/cm^2^) on MAGI1 expression in the BTF system and two additional flow systems: the orbital shaker [20] and the parallel plates (ibidi) [27] models. As in the BTF model, 24 h fluid shear stress in orbital shaker and parallel plates system also induced HUVEC alignment (Appendix A). Interestingly, shear stress induced an increase in levels of MAGI1 in the three systems (Figure 1B). Increased protein levels were detected as soon as 3 h after flow start and were maximal at 24 h (Figure 1C). Flow also increased MAGI1 mRNA levels, indicative of transcriptional regulation (Figure 1D).

The data demonstrate that MAGI1 is present at endothelial cell junctions under both static and physiological shear stress conditions and that shear stress induces MAGI1 mRNA and protein expression. For subsequent biochemical studies, we adopted the orbital shaker model as it allows treating larger amounts of cells in multiple parallel conditions. 

### 3.2. MAGI1 Facilitates Endothelial Cell Alignment and KLF4 Expression in Response to Fluid Shear Stress

As MAGI1 co-localizes with VE-cadherin at cell-cell contacts and VE-cadherin was indicated to be a part of a junctional mechanosensory complex together with PECAM-1 and VEGFR-2 [28], we hypothesized that MAGI1 itself could participate in the mechanotransduction response to fluid shear stress. To test this hypothesis, we monitored the effect of modulating MAGI1 levels on flow induced HUVEC alignment [29]. HUVEC with overexpressed or silenced MAGI1 were exposed to fluid shear stress (10 dyn/cm^2^) for 24 h. MAGI1 overexpression facilitated HUVEC alignment while MAGI1 silencing interfered with it, compared to control cells (Figure 2A). Next, we monitored the effect of MAGI modulation on the expression of Krüppel-like factor 4 (KLF4), a transcription factor regulated by flow that mediates essential vascular functions in response to flow [30]. MAGI1 silencing prevented flow-induced KLF4 expression (Figure 2B), while MAGI1 overexpression increased KLF4 expression in the absence of flow (Figure 2C).

HUVEC stimulation with vascular endothelial growth factors (VEGF) or thrombin, two physiological activators of endothelial cells, did not induce MAGI1 expression, suggesting that the observed induction is a rather specific response to shear stress.

These results demonstrated that endothelial cell MAGI1 is induced by fluid shear stress and is involved in mediating endothelial cell response to it.

### 3.3. MAGI1 Promotes eNOS Phosphorylation and NO Production in Response to Fluid Shear Stress

Fluid shear stress is a potent regulator of eNOS function and NO production [31]. To test the role of MAGI1 in shear stress-mediated eNOS activation, we monitored eNOS phosphorylation at Ser^1177^, a phosphorylation site associated with increased eNOS activity [1,4] in MAGI1-silenced HUVEC or control HUVEC in response to flow (10 dyn/cm^2^ for 24 h). In non-silenced HUVEC, we observed a robust induction of eNOS Ser^1177^ phosphorylation in response to flow while in MAGI1-silenced HUVEC, eNOS Ser^1177^ phosphorylation was strongly reduced (Figure 3A). Conversely, MAGI1 overexpression increased eNOS Ser^1177^ phosphorylation in the absence of flow (Figure 3B). In time course experiments, we observed an initial decrease of eNOS Ser^1177^ phosphorylation upon exposure to flow in non-silenced HUVEC, followed by an increase to maximal levels at 24 h. MAGI1 silencing prevented the flow-induced phosphorylation at later time points (6 and 24 h) (Figure 3C, Appendix A). Consistent with these observations, MAGI1 overexpression enhanced NO production under both static and flow conditions compared to mock and non-silencing controls, while MAGI1 silencing reduced NO production in response to flow (Figure 3D).

These results indicated that MAGI1 mediates shear stress-induced sustained eNOS activation and NO production. 

### 3.4. MAGI1-Induced eNOS Phosphorylation at Ser^1177^ is Independent of AKT and CaMKII

MAGI1 guanylate kinase domain has no enzymatic activity, a feature shared with all MAGUK proteins [15], implying that MAGI1 phosphorylates eNOS indirectly through other kinases. Four kinases activate eNOS through phosphorylation at Ser^1177^: AKT, CaMKII, PKA and AMPK [1,4]. AKT was a prime candidate as it phosphorylates eNOS in response to shear stress [32] and the tumour suppressor protein PTEN, a negative regulator of AKT, binds to MAGI1 [33]. MAGI1 silencing and overexpression, however, had no impact on AKT Ser^473^ phosphorylation (Appendix A). Also, the AKT inhibitor MK2206, while effectively suppressing basal AKT Ser^473^ phosphorylation, had no effect on eNOS Ser^1177^ phosphorylation induced by MAGI1 overexpression (Appendix A).

Next we looked at CaMKII, an alternative candidate as MAGI1 interacts with α-actinin-4 [34] and α-actinin-4 itself contributes to CaMKII activation, and binds to and regulates eNOS activity [35,36]. To this end, we monitored the effect of the CaMKII kinase inhibitor KN-93 on eNOS Ser^1177^ phosphorylation in MAGI1 control and overexpressing HUVEC. MAGI1 overexpression induced a small increase in CAMKII phosphorylation and KN-93 treatment effectively inhibited CAMKII phosphorylation but did not impact eNOS Ser^1177^ phosphorylation (Appendix A).

From these results, we concluded that AKT and CaMKII are not significantly implicated in MAGI1-induced eNOS Ser^1177^ phosphorylation.

### 3.5. AMPK and PKA Mediate MAGI1-Induced eNOS Ser^1177^ Phosphorylation

AMPK was recently shown to regulate eNOS Ser^1177^ phosphorylation independently of AKT [37]. Thus, we examined the effect of MAGI1 on AMPK phosphorylation and whether inhibition of AMPK suppressed eNOS phosphorylation. MAGI1 overexpression induced AMPK and eNOS Ser^1177^ phosphorylation while the AMPK inhibitor dorsomorphin inhibited both of these phosphorylations (Figure 4A). PKA also regulates eNOS phosphorylation at Ser^1177^ in response to shear stress, independently of AKT [38]. MAGI1-overexpression increased PKA phosphorylation and treatment with the PKA inhibitor H89 inhibited PKA and eNOS phosphorylation in control and MAGI1-overexpressing HUVEC (Figure 4B). Time-course experiments showed that eNOS phosphorylation paralleled PKA phosphorylation (Figure 4C). In time course experiments, we observed a clear induction of sustained PKA phosphorylation in response to flow in NS HUVEC, while phosphorylation was blunted in MAGI1-silenced HUVEC (Figure 4D, Appendix A). Consistent with these results, treatment with H89 suppressed NO production in wild type HUVEC under static and fluid shear stress conditions (Figure 4E), confirming the role of PKA in regulating eNOS activity independent of AKT.

From these experiments, we conclude that MAGI1 regulates eNOS Ser^1177^ phosphorylation and activity through PKA and AMPK.

### 3.6. Transgenic Endothelial Cell Expression of MAGI1 Induces eNOS Phosphorylation at Ser^1177^ and Increases NO Production in Endothelial Cells

Next, we proceeded to collect evidence that MAGI1 activates eNOS in endothelial cells in vivo. To this end, we first generated a transgenic mouse line harbouring human MAGI1 cDNA under the tetOS promoter response element (ST, single transgenic, tetOS::MAGI1) and crossed with transgenic mice expressing the tet transactivator (tTA) under the control of the endothelial cell specific VE-cadherin promoter [39] to obtain a double transgenic line (DT, VEC:tTA::tetOS:MAGI1) with doxycycline repressible expression of MAGI1 in endothelial cells (Appendix A). In the absence of doxycycline mice constitutively overexpress transgenic MAGI1 in endothelial cells, while in the presence of doxycycline transgenic expression of MAGI1 is silenced. MAGI1 mRNA and protein expression analysis in total lung lysates from ST and DT of control mice or mice treated for three days with doxycycline, confirmed that transgenic (human) MAGI1 mRNA and MAGI1 protein were only expressed in DT mice in the absence of doxycycline (Appendix A). The weak protein signal visible in ST and DT+doxycycline conditions is due to the levels of endogenous MAGI1 detected by the anti-MAGI1 antibody (Appendix A).

To demonstrate MAGI1 expression in endothelial cells, we isolated lung microvascular endothelial cells (LEC) from ST and DT mice and cultured them for three days in the absence or presence of doxycycline. MAGI1 protein was expressed in DT LEC, but not in DT LEC+doxycycline or ST LEC (Figure 5A). Increased PKA Thr^197^ phosphorylation was visible in LEC of DT mice without doxycycline, consistent with data obtained in HUVEC (Figure 5A). Importantly, increased phosphorylation of eNOS was observed in LEC from DT mice compared to LEC from ST mice (Figure 5B). To collect direct evidence for MAGI1-induced eNOS activation in vivo, we stained lungs for phosphorylated eNOS Ser^1177^. Increased total MAGI1 protein and phospho-eNOS Ser^1177^ staining were visible in lungs of DT mice compared to ST (Figure 5C). Consistent with these observations, production of NO in freshly isolated LEC was significantly increased in LEC from DT mice compared to ST mice (Figure 5D).

Taken together, these results demonstrated that transgenic expression of MAGI1 in endothelial cells in vivo increases PKA and eNOS Ser^1177^ phosphorylation and enhances NO production in lung microvascular endothelial cells. 

## 4. Discussion

Tight regulation of NO production is key to many homeostatic and adaptive responses in the vascular system. Activity of eNOS, the main NOS in endothelial cells, is regulated by several mechanisms, including phosphorylation through AKT-dependent and -independent pathways (Figure 6) [1,7]. The relative importance of these multiple pathways is still unclear, and in particular the regulatory components involved in PKA-dependent phosphorylation of eNOS are not fully elucidated. 

Here we report that the scaffolding protein MAGI1 contributes to PKA- and, to a lesser extent, AMPK-dependent eNOS activation and NO production. This conclusion is based on the following observations: i) Fluid shear stress induced MAGI1 expression in HUVEC in three independent models, silencing of MAGI1 expression impinged on eNOS Ser^1177^ phosphorylation and NO production; ii) MAGI1 overexpression in HUVEC and in endothelial cells of transgenic mice, increased PKA phosphorylation, induced eNOS Ser^1177^ phosphorylation in vivo, in vitro and ex vivo, as well as NO production in vitro and ex vivo. iii) PKA inhibition suppressed eNOS Ser^1177^ phosphorylation and NO production. iv) MAGI1 also induced AMPK Thr^172^ phosphorylation, and its inhibition partially reduced MAGI-induced eNOS phosphorylation. 

A number of MAGI1 physiological functions and involvements in pathologies, including cancer, have been reported, but only little is known on the putative function of MAGI1 in vascular biology. At the cellular level, MAGI1 plays an important role in stabilizing cell-cell contacts in epithelial cells, through tight junctions [15]. MAGI1 regulates apoptosis [40], synaptic development, plasticity and function [41]. In endothelial cells, MAGI1 enhances VE-cadherin-mediated cell adhesion [18] and recruits ESAM to cell contacts [42]. In addition to the modulation of PI3-K/PTEN/AKT [43], Wnt/β-catenin [22] and IQGAP signalling [44], there is a paucity of information on intracellular signalling events modulated by MAGI1. Our results point towards a novel function of endothelial cell MAGI1, namely the regulation of eNOS phosphorylation and activity in response to shear stress through the PKA and AMPK pathways. MAGI1 co-localizes at cell-cell junctions together with VE-cadherin, a sub-cellular location where eNOS and the catalytic subunit of PKA were reported in conjunction with the regulation of NO production [45]. How MAGI1 activates PKA is unclear at this point. PKA is a cyclic adenosine monophosphate (cAMP)-dependent kinase and two cAMP molecules bind to each PKA regulatory subunit causing a conformational change activating its catalytic subunit [46]. PKA interacts with RAPGEF2 (PDZ-GEF1) required for cAMP-dependent downstream signalling [47]. Intriguingly, MAGI1 has been shown to interact with PDZ-GEF1 in vascular endothelial cells where it is required for VE-Cadherin dependent Rap1 activation [47]. It is plausible that MAGI1 activates PKA through a scaffolding effect clustering RAPGEF2 and PKA at adherens junctions and thereby facilitating cAMP-mediated PKA activation.

Further evidence for a potential contribution of MAGI1 to endothelial physiology is corroborated by its involvement in the induction of KLF4 expression in response to flow. There are numerous studies showing that laminar shear stress is inducing KLF4 expression through both transcriptional [48] and epigenetic [49] regulation leading to increased eNOS synthesis and NO production. In contrast, disturbed flow increases DNA methylation of GpC islands in the KLF4 promoter, leading to its transcriptional suppression [49]. KLF4 modulates the progression of multiple vascular conditions by regulating the transcription of several genes in endothelial cells, smooth muscle cells and inflammatory cells [50]. For example, KLF4 reduces smooth muscle cell differentiation and proliferation by interacting with the serum response factor ELK1, histone deacetylases and p53 [50] and represses vascular inflammation by inhibiting NF-κB [51].

In view of the contribution of KLF4 in mediating responses to flow and its multiple roles in vascular physiology and pathology, it will be essential to unravel how MAGI1 regulates KFL4 expression, and in particular whether KLF4, in turn, may regulate MAGI1 expression. Some of the ongoing experiments are specifically addressing this important question.

Considering the importance of NO metabolism in many cardiovascular diseases, in particular atherosclerosis and hypertension [5,52], it will be important to assess whether endothelial MAGI1 function or dysfunction may be involved in the genesis or progression of such pathologies.

## 5. Conclusions

The presented study demonstrates for the first time that the scaffolding protein MAGI1 regulates shear stress-mediated NO production through a PKA/AMPK dependent eNOS phosphorylation in endothelial cells. These results add a previously unrecognized element to the characterization of MAGI1 function in vascular biology and to the complex mechanisms regulating NO production in endothelial cells.

## Figures and Tables

**Figure 1 cells-08-00388-f001:**
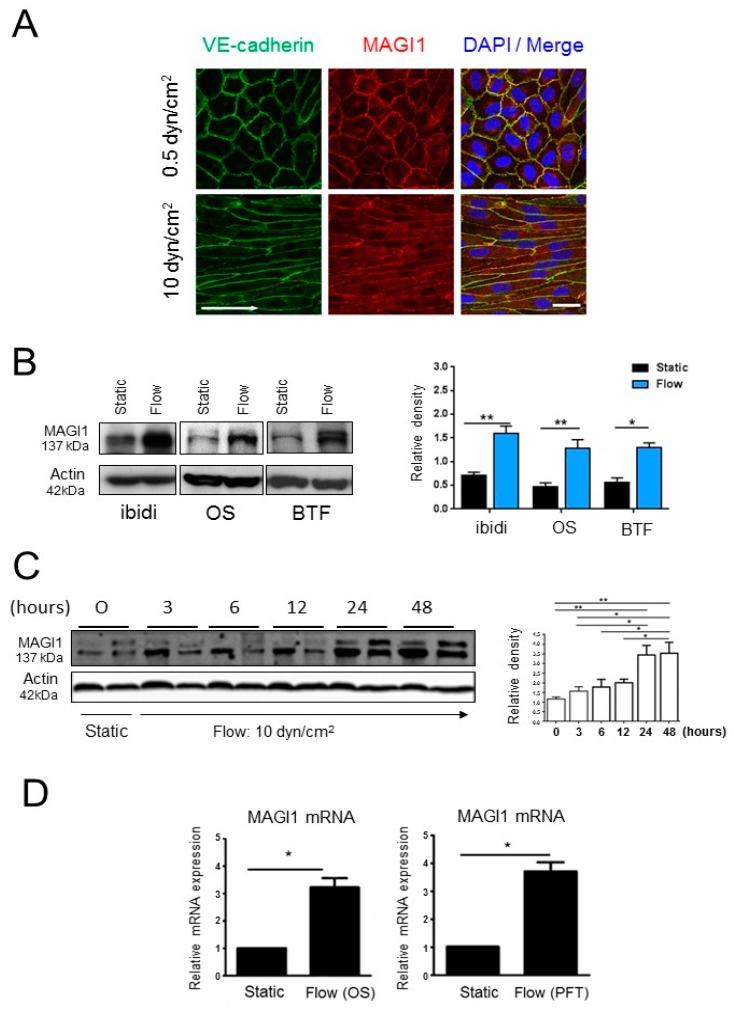
MAGI1 localizes at endothelial cell junctions and its expression is induced by shear stress. (**A**) Confocal laser microscopy of MAGI1 and VE-cadherin-stained HUVEC confluent cultures under static conditions (0.5 dyn/cm^2^) and after enhanced fluid shear stress of 10 dyn/cm^2^ for 24 h using the BioTechFlow system (BTF). MAGI1 co-localizes with VE-cadherin under static and flow conditions. Arrows: direction of flow. N = 5. Scale bar = 25 μm. (**B**) HUVEC cultured under static conditions or exposed to 10 dyn/cm^2^ fluid shear stress (flow) in the parallel plates (ibidi), orbital shaker (OS) and cone-and-plate (BTF) systems were analysed by Western blotting for MAGI1 protein levels. Fluid shear stress induced MAGI1 protein in the three models. N = 3–5. (**C**) HUVEC cultured under static conditions (0) or exposed to 10 dyn/cm^2^ fluid shear stress in the orbital shaker (OS) systems for the indicated time were analysed for MAGI1 protein levels by Western blotting. N = 2, duplicate samples each condition. Maximal MAGI1 induction is reached at 24 h. Beta actin blotting are shown as loading controls. Relative band quantifications by densitometry of multiple experiments are shown next to the blots. (**D**) MAGI1 mRNA levels in HUVEC under static and fluid shear stress conditions in the OS and BTF models measured by real time RT-PCR. Fluid shear stress induces MAGI1 mRNA expression. N = 3, in triplicate conditions. * *p* < 0.05; ** *p* < 0.01.

**Figure 2 cells-08-00388-f002:**
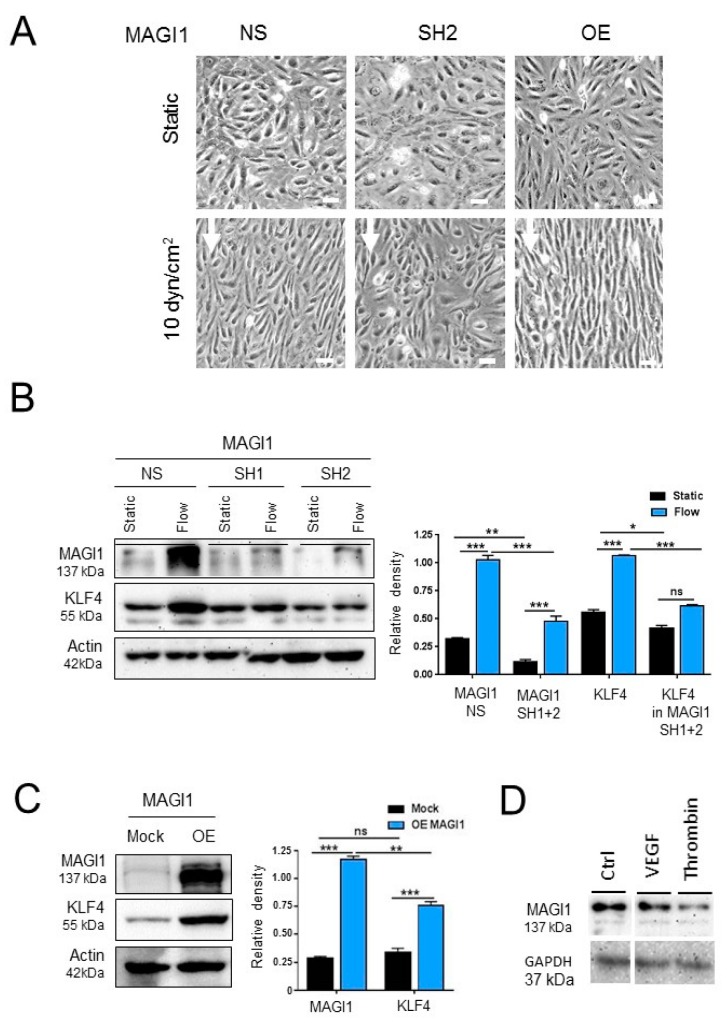
MAGI1 mediates endothelial cell alignment and KLF4 expression upon exposure to fluid shear stress. (**A**) Control HUVEC (NS), HUVEC with silenced (SH) or overexpressed (OE) MAGI1 were exposed to laminar shear stress (10 dyn/cm^2^) in the orbital shaker model (OS) for 24 h. MAGI1 SH perturbs, while MAGI1 OE facilitates alignment. Arrows: direction of flow. N = 5. Scale bar = 20 µm. (**B**) Western blotting analysis of control (NS) and MAGI1 silenced (SH1 and SH2) HUVEC under static and orbital flow conditions (OS) for 24 h. MAGI1 silencing prevents KLF4 induction under shear stress. N = 3. (**C**) Western blotting analysis of control (Mock) and MAGI1-overexpressing (OE) HUVEC under static conditions. N = 3. (**D**) Western blotting analysis of cultured control HUVEC (Ctrl) or HUVEC exposed to VEGF (100 ng/mL for 24 h) and thrombin (10 U/mL for 30 min). Beta actin or GAPDH are shown as loading controls. Relative band quantifications by densitometry of multiple experiments are shown next to the blots. * *p* < 0.05; ** *p* < 0.01; *** *p* < 0.001.

**Figure 3 cells-08-00388-f003:**
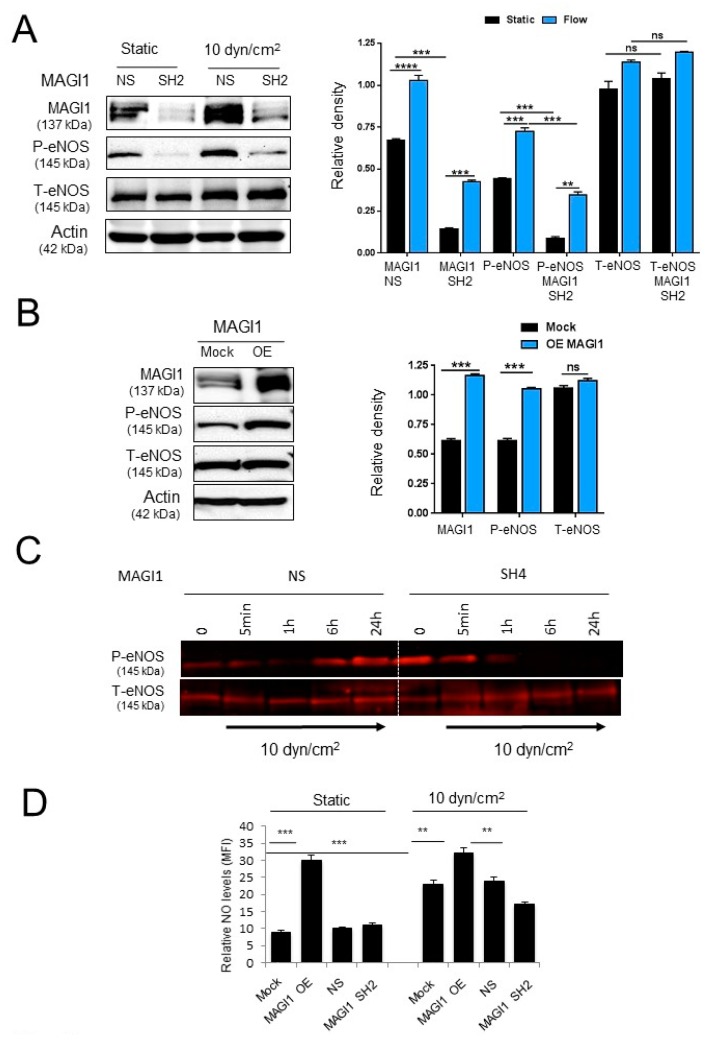
MAGI1 promotes eNOS phosphorylation. (**A**) Control (NS) and MAGI1-silenced (SH) HUVEC under static or flow (10 dyn/cm^2^, OS) conditions for 24 h were analysed by Western blotting for eNOS Ser^1177^ phosphorylation, total eNOS and MAGI1. No. of repetitions = 2. (**B**) Western blotting analysis of eNOS Ser^1177^ phosphorylation total eNOS and MAGI1 in control (Mock) and MAGI1-overexpressing (OE) HUVEC under static conditions. MAGI is required for and mediates eNOS phosphorylation. N = 3. Beta actin is shown as loading control. Relative band quantifications by densitometry of multiple experiments are shown next to the blots. (**C**) Non-silenced (NS) or MAGI1 silenced (SH4) HUVEC were cultured under static conditions (0) or exposed to 10 dyn/cm^2^ fluid shear stress in the orbital shaker (OS) systems for the indicated time and analysed for phosphorylated (Ser1177) and total eNOS levels by Western blotting. MAGI silencing reduces sustained flow-induced eNOS phosphorylation. (**D**) Control HUVEC (Mock and NS), HUVEC with silenced (SH) or overexpressed MAGI1 (OE) under static of flow (10 dyn/cm^2^, 24 h) conditions were monitored for NO production. N = 3, triplicate conditions each. ** *p* < 0.01; *** *p* < 0.001; ns, non-significant.

**Figure 4 cells-08-00388-f004:**
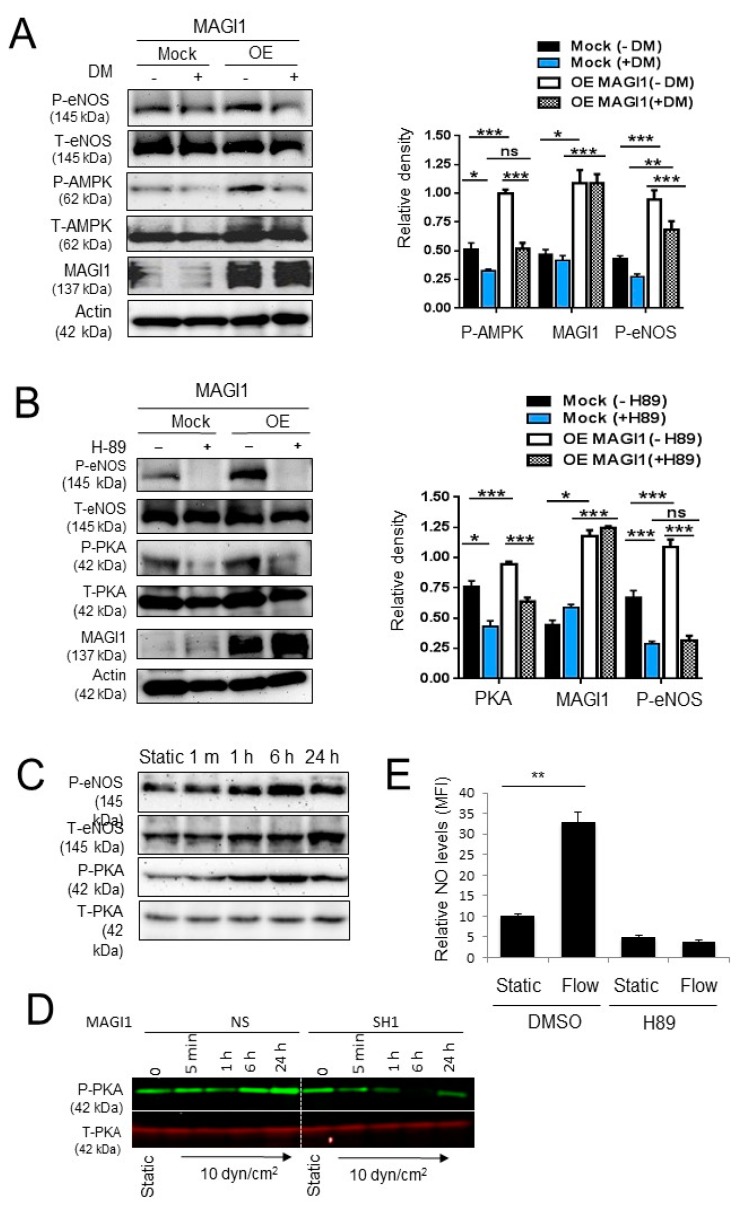
AMPK and PKA mediate MAGI1-induced eNOS phosphorylation. (**A**) Control (Mock) and MAGI1-overexpressing (OE) HUVEC treated with the AMPK inhibitor dorsomorphin (DM, +) or DMSO vehicle control (−), were analysed by Western blotting for phosphorylated AMPK (Thr^286^), total AMPK, phosphorylated eNOS (Ser1177) and total eNOS. N = 3. AMPK mediate MAGI1-induced eNOS phosphorylation. (**B**) Western blotting analysis of phosphorylated PKA (Thr^197^), total PKA, phosphorylated eNOS and total eNOS in control (Mock) and MAGI1 overexpressing (OE) HUVEC in the presence of absence of the PKA inhibitor H-89. PKA mediates MAGI1-induced eNOS phosphorylation. N = 3. Beta actin is shown as loading control. Relative band quantifications by densitometry of multiple experiments is shown next to the blots. * *p* < 0.05; ** *p* < 0.01; *** *p* < 0.001; ns, non-significant. (**C**) HUVEC cultured under static conditions (0) or exposed to 10 dyn/cm^2^ fluid shear stress in the orbital shaker (OS) systems for the indicated time were analysed for phosphorylated eNOS (Ser1177), total eNOS, phosphorylated PKA (Thr^197^) and total PKA levels by Western blotting. (**D**) HUVEC non-silenced (NS) or silenced (SH4) for MAGI1 were cultured under static conditions or exposed to 10 dyn/cm^2^ fluid shear stress in the orbital shaker (OS) systems for the indicated time and analysed for phosphorylated and total PKA levels by Western blotting. MAGI silencing reduces flow-induced sustained PKA phosphorylation. (**E**) HUVEC under static of flow (10 dyn/cm^2^, 24 h) conditions in the presence of vehicle (DMSO) or the PKA inhibitor (H89) were analysed for NO production. H89 treatment inhibits flow induced NO. NO was measured by Nitrixyte™ staining and flow cytometry analysis. N = 3, triplicate conditions each. MFI, mean fluorescence intensity. **, *p* < 0.01.

**Figure 5 cells-08-00388-f005:**
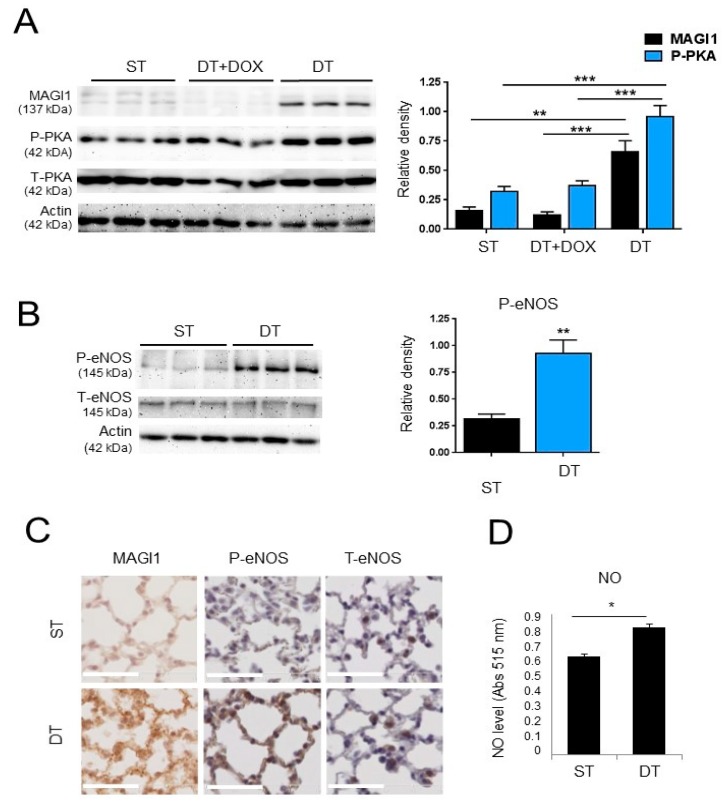
Transgenic expression of MAGI1 in endothelial cells promotes phosphorylation of PKA and eNOS and stimulates NO production. (**A**) Western blotting analysis of total and phosphorylated PKA (Thr^197^) in lung endothelial cells isolated from ST and DT transgenic MAGI1 mice cultured for 3 days in the absence or presence of doxycycline (dox) as indicated. Doxycycline represses MAGI1 expression and PKA phosphorylation. Triplicate mice, N = 3. (**B**) Western blotting analysis of lung endothelial cells isolated from ST and DT mice for phosphorylated Ser^1177^ eNOS (P-eNOS) and total eNOS (T-eNOS). DT lung endothelial cells have elevated eNOS phosphorylation. N = 2, triplicate mice (six independent cultures). Beta actin is shown as loading control. Relative band quantifications by densitometry of multiple experiments are shown next to the blots. (**C**) Lung sections from ST and DT MAGI1 mice were stained for MAGI1, phosphorylated Ser^1177^ eNOS (P-eNOS) or total eNOS (T-eNOS). Increased P-eNOS staining (brown) is visible in DT mice. Scale bar = 50 µm. (**D**) Analysis of NO production by DAF-2DA staining and fluorimetric measurement in lung endothelial cells isolated from. N = 3, triplicate conditions each. ST, single transgenic mice; DT, double transgenic mice. * *p* < 0.05; ** *p* < 0.01; *** *p* < 0.001.

**Figure 6 cells-08-00388-f006:**
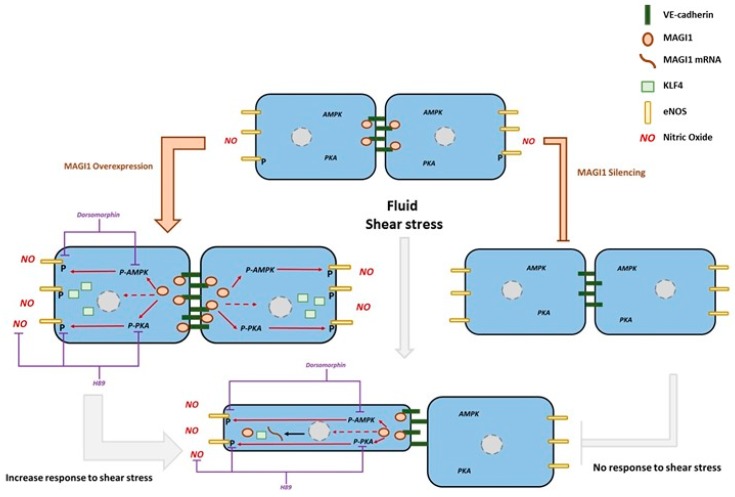
Proposed model. MAGI1 localizes at cell-cell contacts with VE-Cadherin. Shear stress induces MAGI1 expression, which promotes endothelial cell alignment to flow, KLF4 expression, Ser^1177^eNOS phosphorylation and NO production. MAGI1 induced Ser^1177^eNOS is mainly mediated through PKA and to a lesser extent AMPK. AKT or CAMKII, two additional kinases also reported to phosphorylate and activate eNOS, are not involved in MAGI1-mediated Ser^1177^eNOS phosphorylation.

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
