# Peer review of "MAGI1 Mediates eNOS Activation and NO Production in Endothelial Cells in Response to Fluid Shear Stress"

_cells, 2019, doi:10.3390/cells8050388_

Round 1

Reviewer 1 Report

The study has been well designed and the results are convincing. The presentation can be improved as described below.

General comments:

The authors aim to determine the role of MAGI1 in shear
stress induced eNOS and NO upregulation in vascular endothelial cells (EC).
They found the co-localization of MAGI1 and VE cadherin in EC using co-focal
microscopy. Shear stress increases the expression of MAGI1 in EC. MAGI1 shRNA
inhibited shear stress induced eNOS and NO upregulation. Overexpression of
MAGI1 caused opposite effects. PKA and AMPK phosphorylation of eNOS is
implicated in this process. The results have been verified in animal studies.
The design of the studies appears to be appropriate and the results are
consistent with the conclusion. The outcome contributes to the underlying
mechanism for shear stress induced eNOS and NO generation from vasculature.

Specific comments:

Fig.1C and Fig.3A. The two experiments were
described as N=2. Does the statistical analyses are based on n=2/group.

Methods: What age and sex of mice were used in the
study?

Results, line 295: Fig.2D should be added  at the end of the paragraph

Line 305: Phosphorylation of eNOS was detected in
EC transfected with MAGI1 under shear stress for 5 min but not in EC exposed to
shear stress for ≥1h. How long after the start of the gene silence of MAGI1,
the shear stress started?

Results 3.6: the rationale for using doxycycline
has not been described in results or discussion.

Discussion appears to be weak. The scheme is
helpful, but the mechanism has not been well elucidated in the text. For
example: how overexpression of MAGI1 triggers PKA and AMPK to phosphorylate
eNOS.

A number of styles errors cross the manuscript,
for examples:

Abstract: no full terms for VE and MAGUK.

Introduction: no full term for PDZ or TRIP6.

Methods , line 106: no full term for HUVEC ,
however, the full term first time appeared in results at line 235.

Line 230: “***= p<0.0001” should be ****?

Line 301: “In non silence, HUVEC we observed”,
the sentence should be reorganized.

Fig. 4C, label for P-eNOS (145 Ka) dislocalized.

Author Response

Q1. Fig.1C and Fig.3A. The two experiments were described as N=2. Does the statistical analyses are based on n=2/group. Methods: What age and sex of mice were used in the study?

R1. In these figures, N represents the no. of times the experiment was repeated. So, Figure 1C is representative of an experiment repeated twice (N=2), with 2 samples each (=2 lanes per WB). In total we had results from 4 samples that were used for statistical analysis after densitometry measurement. Each sample (=lane) is the pool of 3 independent culture wells (6-well plates).

Figure 3A, the experiment was repeated twice (N=2). Here again each sample (=lane) is the pool of 3 independent culture wells. So, one lane represents the average of 4 independent culture wells (6-well plates). This information was missing from the materials and methods section and was added under 2.8.

We used both male and female mice between 6 and 10 weeks of age. This information was missing and has been stated in the revised MM section 2.11.

Q2. Line 230: “***= p<0.0001” should be ****?

R2. Yes, indeed. Thank you for spotting this mistake, it has been corrected in the revised manuscript.

Q3. Line 305: Phosphorylation of eNOS was detected in EC transfected with MAGI1 under shear stress for 5 min but not in EC exposed to shear stress for ≥1h. How long after the start of the gene silence of MAGI1, the shear stress started?

R3. HUVEC are constitutively silenced for MAGI1 by infecting the cells with LV expressing MAGI1 cDNA, and selected with Puromycin for 48 hours and used usually in the following 7-10 days (2-3 passages). As MAGI1 induced eNOS phosphorylation, this means that these cells have increased eNOS phosphorylation independently of flow, due to overexpressed MAGI1. In WT HUVEC, flow induces a first rapid eNOS phosphorylation, followed by a decrease and then a progressive increase peeking at 24 hours. Thus, the flow-independent (in MAGI1 overexpressing cells) and flow-dependent eNOS phosphorylation explain this apparent discordance.

Q4. Results 3.6: the rationale for using doxycycline has not been described in results or discussion.

R4. Double transgenic mice allowing regulated MAGI1 expression were generated in a two-step approach. Firstly we inserted human MAGI1 cDNA (NM_004742.2, 3770bp) downstream of the pTetOS responder element in vector then used to generate transgenic mice. Secondly, these mice were crossed with a transgenic line expressing the tet transactivator (tTA) under the control of the endothelial cell specific VE-cadherin promoter to generate double transgenic (DT) mice. The DT mice harbor MAGI1 ORF under control of the pTetOS responder element activated by tTA constitutively expressed in endothelial cells only by the VE-cadherin promoter.

Doxycycline was used to regulate MAGI1 expression: in the absence of doxycycline DT mice constitutively expressed MAGI1 in endothelial cells (TetOFF), and addition of doxycycline blocks tTA binding to tetOS preventing MAGI1 expression. Thus, in the absence of tetracyclic, mice constitutively overexpress transgenic MAGI1 in endothelial cells, while in the presence of tetracyclic transgenic expression of MAGI1 is silenced. To improve clarity, we have added this last sentence in the results section, at 3.6 and in the MM section at 2.11 Animal studies.

Q5. Discussion appears to be weak. The scheme is helpful, but the mechanism has not been well elucidated in the text. For example: how overexpression of MAGI1 triggers PKA and AMPK to phosphorylate eNOS.

R5. This is one of the interesting questions raised by our study for which we have no answer. For the time being we have only hypotheses and dedicated experiments will be necessary to characterize this mechanism. We have addressed this point in the discussion by adding the following sentence:

How MAGI1 activates PKA is unclear at this point. PKA is a cyclic adenosine monophosphate(cAMP)-dependent kinase as two cAMP molecules bind to each PKA regulatory subunit causing a conformational change activating its catalytic subunit. PKA interacts with RAPGEF2 (PDZ-GEF1) required for cAMP-dependent downstream signalling. Intriguingly, MAGI-1 has been shown to interact with PDZ-GEF1 in vascular endothelial cells where it is required for VE-Cadherin dependent Rap1 activation. It is plausible that MAGI1 activates PKA through a scaffolding effect clustering RAPGEF2 and PKA at adherens junctions and thereby facilitating cAMP-mediated PKA activation.

Q6. A number of styles errors cross the manuscript, for examples:

Abstract: no full terms for VE and MAGUK.

Introduction: no full term for PDZ or TRIP6.

Methods , line 106: no full term for HUVEC , however, the full term first time appeared in results at line 235.

Line 230: “***= p<0.0001” should be ****?  

Line 301:  “In non silence, HUVEC we observed”, the sentence should be reorganized.

Fig. 4C, label for P-eNOS (145 Ka) dislocalized.

R6. Thank you for these comments that we have addressed in full and relevant corrections have been made to the revised text.

Full-form for VE, MAGUK, TRIP6, PDZ, WW and HUVEC have been added. We also explained KLF.

Line 301 seems fine but searching for ‘’in non silence’’ led us to Line 340 where the sentence has been reorganized.

Fig 4C label has been localized back.

Reviewer 2 Report

Authors studied the role of MAGI1 in response to fluid shear stress in endothelial cell. You demonstrated that MAGI1 in response to fluid shear stress-induced PKA/AMPK dependent eNOS activation and NO production. The paper is well written and findings are well documented, still minor revision is required. 

1. Line 25, MAGI1 silencing perturbed flow depended responses  --> MAGI1 silencing perturbed flow depended on responses

2.Line 29, Endothelial cell specific transgenic expression --> Endothelial cell-specific transgenic expression

3. Line 48, Regulation of eNOS is complex, and occurs --> Regulation of eNOS is complex and occurs 

4. Line 236, MAGI1 localized at cell-cell contact as a continuous staining  --> MAGI1 localized at cell-cell contact as continuous staining 

Author Response

Q1. Line 25, MAGI1 silencing perturbed flow depended responses  --> MAGI1 silencing perturbed flow depended on responses

R1. We have corrected as : MAGI1 silencing perturbed flow-dependent responses,…

Q2.Line 29, Endothelial cell specific transgenic expression --> Endothelial cell-specific transgenic expression

R2. We have corrected as suggested.

Q3. Line 48, Regulation of eNOS is complex, and occurs --> Regulation of eNOS is complex and occurs 

R3. We have corrected as suggested.

Q4. Line 236, MAGI1 localized at cell-cell contact as a continuous staining  --> MAGI1 localized at cell-cell contact as continuous staining 

R4. We have corrected as suggested.

We thank this reviewer for these suggestions.

Reviewer 3 Report

Ghimire and colleagues discovered a role for MAGI1 as a specific mediator of shear stress response in endothelial cells. Indeed, they characterized two molecular pathways, involving the transcription factor, KLF4, and the PKA- and AMPK-dependent phosphorylation of endothelial nitric oxide synthase (eNOS) in aligned endothelial cells in which MAGI1 expression was silenced or overexpressed. The manuscript is well-written and the presented experiments support an additional contribution to the mechanisms of NO production in endothelial cells, that may have future impact in pathological settings, such as cardiovascular diseases and the crosstalk between cancer and its environment. Yet, some issues need to be better clarified and characterized.

1. One weakness is the lack of a molecular link between KLF4 transcription factor increased expression and the change in eNOS phosphorylation in response to MAGI1 expression changes. Since, NO-related genes are among those targeted by KLF4 promoter, I would expect a crosstalk between the two molecular pathways in response to shear stress. For example, further experiments showing whether the KLF4 promoter undergoes any modifications in response to shear stress or NO production would provide a complete framework to the manuscript. On the other hand, it would be advisable to check eNOS phosphorylation after KLF4-promoter negative modulation to assess whether the two cellular responses have an additional or reciprocal inhibition effect.

2. The crosstalk between MAGI1 and VE-Cadherin at the cell-to-cell-contacts should be better characterized. Does VE-cadherin expression increases, together with MAGI1 one, in response to shear stress (Figure 1)? Moreover, is it indeed possible that MAGI1 and VE-cadherin bind each other at the cell-to-cell-contacts?

Author Response

Q1. One weakness is the lack of a molecular link between KLF4 transcription factor increased expression and the change in eNOS phosphorylation in response to MAGI1 expression changes. Since, NO-related genes are among those targeted by KLF4 promoter, I would expect a crosstalk between the two molecular pathways in response to shear stress. For example, further experiments showing whether the KLF4 promoter undergoes any modifications in response to shear stress or NO production would provide a complete framework to the manuscript. On the other hand, it would be advisable to check eNOS phosphorylation after KLF4-promoter negative modulation to assess whether the two cellular responses have an additional or reciprocal inhibition effect.

R1. Thanks for your valuable input. This is indeed one of the outstanding questions we plan to address in future experiments: Does MAGI1 has a role as mechanotransducer in the VEGFR2-VEC-CD31 complex? How does MAGI activate PKA? How does MAGI1 regulate KLF4 expression? Does MAGI expression in endothelial cell modulate NO-depended cardiovascular diseases in vivo.

Addressing each of these questions, including the KLF4 cross talk and the possible involvement of other KLFs, will represent a full study by itself, which is beyond the scope of the current work.

Stimulated by these comments, however, we have started some experiments and we have observed that KLF4 itself regulates MAGI1 expression: KD-KLF4 downregulates MAGI1 and OE-KLF4 upregulates MAGI1.

This suggests a positive cross-talk between MAGI1 and KLF4, whereby MAGI1 acts as part of the mechanotransducer complex that contribute to KLF4 expression, and KLF4 expression induced MAGI1 expression to reinforce response to flow. These data are preliminary and still ongoing, and too early for inclusion into this manuscript.

To emphasize the important of this observation and the need for additional experiments, we have elaborated a new sentence at the end of the discussion on KLF4:

In view of the contribution of KFL4 in mediating responses to flow and its multiple roles in vascular physiology and pathology, it will be essential to unravel how MAGI1 regulates KFL4 expression, and in particular whether KLF4, in turn, may regulate MAGI1 expression. Some of the ongoing experiments are specifically addressing this important question.

Q2. The crosstalk between MAGI1 and VE-Cadherin at the cell-to-cell-contacts should be better characterized. Does VE-cadherin expression increase, together with MAGI1 one, in response to shear stress (Figure 1)? Moreover, is it indeed possible that MAGI1 and VE-cadherin bind each other at the cell-to-cell-contacts?

R2. The impact of flow and shear stress on VE cadherin has been largely characterized in the literature.

Shear stress induces a redistribution of VE cadherin at cell-cell contacts, from a continuous linear distribution to a continuous but interdigitated one, reflecting the remodeling of cell-cell adhesion during shear stress. MAGI1 staining parallels VE cadherin localization under both static and flow conditions. We have shown this in Fig. 1A of the manuscript and also in the figure below.

However, in contrast to the strong and progressive increase of MAGI1 expression both at protein and mRNA levels in response to flow described for the first time here, VE cadherin expression is known from the literature to be modulated by flow in two phases: first a down regulation (4-8 hours after flow initiation) followed by an upregulation. The upregulation, however is in general mild [1-4]. The main responses of VE cadherin to flow are increased phosphorylation, increased association with the cytoskeleton and the cell membrane, and strengthened adherens junctions. We felt that monitoring VE cadherin expression would not add relevant information to this study and has been extensively characterized before.

The question of whether MAGI1 and VE-cadherin bind each other at the cell-to-cell-contacts is a highly relevant one. We have circumstantial evidence that VE cadherin and MAGI1 colocalized in a complex, but so far no experimental evidence that MAGI1 interacts directly with VE cadherin. It has been reported that MAGI1 interacts with beta catenin, which is a binding partner of VE cadherin that localizes at adherens complexes. This is again one of the important questions we have to address in future studies (see above), in order to unravel the molecular mechanism of MAGI1 contribution to mechanosensing by the VE cadherin-VEGFR-2-CD31 complex [5].

Round 2

Reviewer 3 Report

I do not have any additional comment.